# Embodied Concept Learner:
# Self-supervised Learning of Concepts and Mapping through Instruction Following

**Mingyu Ding**
HKU

**Yan Xu**
CUHK

**Zhenfang Chen**
MIT-IBM Watson AI Lab

**David Cox**
MIT-IBM Watson AI Lab

**Ping Luo**
HKU

**Joshua B. Tenenbaum**
MIT BCS, CBMM, CSAIL

**Chuang Gan**
UMass Amherst
MIT-IBM Watson AI Lab

**Abstract:** Humans, even at a very early age, can learn visual concepts and understand geometry and layout through active interaction with the environment, and generalize their compositions to complete tasks described by natural languages in novel scenes. To mimic such capability, we propose Embodied Concept Learner (ECL) [1] in an interactive 3D environment. Specifically, a robot agent can ground visual concepts, build semantic maps and plan actions to complete tasks by learning from human demonstrations and language instructions. ECL consists of: (i) an instruction parser that translates the natural languages into executable programs; (ii) an embodied concept learner that grounds visual concepts based on language descriptions/embeddings and a pretrained object proposal network; (iii) a map constructor that estimates depth and constructs semantic maps by leveraging the learned concepts; and (iv) a program executor with deterministic policies to execute each program. ECL has several appealing benefits thanks to its modularized design. Firstly, it enables the robotic agent to learn semantics and depth unsupervisedly acting like babies, *e.g.*, ground concepts through active interaction and perceive depth by disparities when moving forward. Secondly, ECL is fully transparent and step-by-step interpretable in long-term planning. Thirdly, ECL could be beneficial for the embodied instruction following (EIF), outperforming previous works on the ALFRED benchmark when the semantic label is not provided. Also, the learned concept can be reused for other downstream tasks, such as reasoning of object states.

**Keywords:** Embodied AI, Embodied Concept Learner, Instruction Following

## 1 Introduction

Embodied instruction following (EIF) [1] is a popular task in robot learning. Given some multi-modal demonstrations (natural language and egocentric vision, as shown in Fig. 1) in a 3D environment, a robot is required to complete novel compositional instructions in unseen scenes. The task is challenging because it requires accurate 3D scene understanding and semantic mapping, visual navigation, and object interaction.

Recent works for EIF can be typically divided into two streams and they have certain limitations. 1) End-to-end imitation learning methods [1, 2, 3, 4] directly input the visual observation of the current step and language instructions into the model, and output the action for the next step. For example, Pashevich et al. [4] has presented the episodic transformer to predict the agent's actions with

---

[1]Project page: http://ecl.csail.mit.edu/

6th Conference on Robot Learning (CoRL 2022), Auckland, New Zealand.

an attention mechanism and a progress monitor. Such models work by simply memorizing training scenes and trajectories. While they achieve good performance in seen environments, they fail to generalize well in unseen scenes. Furthermore, these black-box models often lack transparency, interpretability, and generalizability. 2) Mapping-based methods [5, 6] leverage the map representations [7, 8, 9, 10] by building a 3D voxel map from the predicted depths and instance segmentation masks. A semantic top-down map of the scene is then constructed and updated at each step. These works perform explicit exploration and interactions through semantic search policies [6] to achieve the natural language goal, which is transparent and interpretable. However, they assume that the agent has learned the depth and semantics passively from large amounts of data. The semantic labels and depth supervision are often labor-intensive and hard to obtain in the real world. We argue that such supervision signals are unnecessary since we can learn language concepts and visual disparity through interactions in the environments. For example, by achieving the goal described in Fig. 1, humans can learn what the concepts "knife" and "table" are and perceive that the table in frame 2 is physically closer to the agent than frame 1.

This paper answers a question naturally raised from the above issues: can we make the agent behave like a baby? A baby is able to learn domain knowledge from environmental interactions and expert demonstrations without additional supervision to achieve the natural language goal. We speculate that babies do this possibly in a way similar as: (i) Learn skills and concepts from expert demonstrations (environment observations and language instructions), *e.g.*, the skill "place" and concepts "knife" and "table" can be grounded from the demonstra-

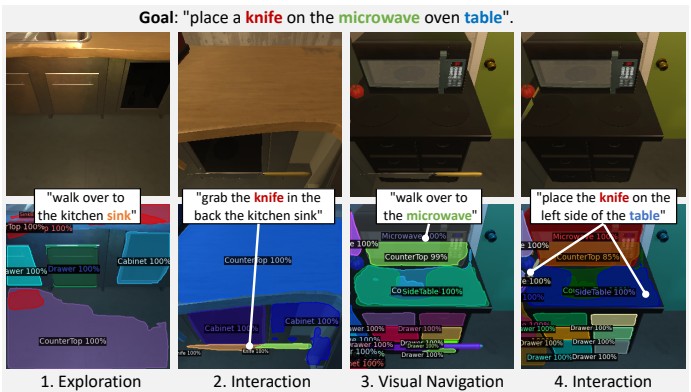

Figure 1: An example of a language goal and its corresponding four subgoals. The top and bottom rows show visual observations by the robotic agent and our grounded semantics, respectively. We show that we align object concepts encoded in subgoals with visual proposals to learn concepts in the embodied environment.

tion "place a knife on the microwave oven table". (ii) Given a new compositional language goal like "put a clean tomato on the dining table" in Fig. 2, one may process it into many subgoals, like "pickup tomato", "clean tomato", and "put it on the table". (iii) Explore the scene and build a semantic map, where depth information is estimated automatically based on the disparity when moving forward or backward. (iv) Complete each subgoal based on the learned semantic map and skills, and update the semantic map dynamically.

Motivated by the above observations, we propose Embodied Concept Learner (ECL) that mimics baby learning for embodied instruction following. It consists of: (i) an instruction parser that parses the natural languages into executable programs; (ii) an embodied concept learner that aligns language concepts with visual instances (a pretrained object proposal network and a word embedding model are used); (iii) a map constructor based on the grounded semantic concepts and unsupervised depth estimation; and (iv) a program executor with deterministic policies to perform each subtask.

Our contributions are three-fold. 1) We introduce ECL, a modular framework that can ground visual concepts, build semantic maps and plan actions to complete complex tasks by learning purely from human demonstrations and language instructions. 2) ECL achieves competitive performance without semantic labels on embodied instruction following (ALFRED) [1], while maintaining high transparency and step-by-step interpretability. 3) We could also transfer the learned concepts to other tasks in the embodied environment, like the reasoning of object states.

## 2 Related Work

**Embodied Instruction Following.** Language-guided embodied tasks have drawn much attention, including visual language navigation (VLN) [11, 12, 13, 14, 15, 16, 17], embodied instruction following (EIF) [18, 19, 20, 21, 22, 1], object goal navigation [23, 24, 25], embodied question answering [26, 27], program sketch generation [28, 29], and embodied representation learning [30, 31, 32]. Among them, EIF is one of the most challenging tasks, requiring simultaneous accurate 3D scene understanding and memory, visual navigation, and object interaction. [1, 4] present end-to-end models with an attention mechanism to process language and visual input and past trajectories, predicting the subsequent action directly. After that, works [20, 22, 19] modularly process raw language and visual inputs into structured forms by object detectors [33, 34, 35]. The above methods lack transparency and generalizability to unseen scenes. Recently, [5, 6] proposed mapping-based methods to convert visual semantics and estimated depth into Bird's-eye-view (BEV) semantic maps and navigate based on the spatial memory. However, such methods require depth and semantic supervisions, hence impractical in real-world scenarios.

**Visual Grounding and Concept Learning.** Our work is also related to visual grounding [36, 37, 38, 39, 40, 41, 42, 43] and concept learning [44, 45, 46, 47, 48, 49], which align concepts onto objects in the visual scenes. Traditional visual grounding methods [39, 37] map text phrases and regional features of images into a common space for cross-modality matching. Recently, there are some works [44, 45, 50] learning visual concepts through question answering in passive images or videos. Differently, we study learning both visual concepts and physical depths through language instructions in the active embodied environment, which is more similar to how humans learn in the real world. Some works study language grounding in 3D world [51, 52, 53]. However, they do not involve robot agents and active exploration. Hermann et al. [48] interprets language in a simple simulated 3D environment, which does not consider diverse objects and actions in challenging photorealistic environments.

## 3 Method

In this work, we focus on the embodied instruction following task, *i.e.*, a robotic agent is required to achieve the goal in the language instruction by exploring, navigating, and interacting with the embodied environment. Embodied Concept Learner (ECL) includes an instruction parser, an embodied concept learner, a map constructor, and a program executor. The modularized design ensures its transparency and step-by-step interpretability. An overview of ECL is shown in Fig. 2.

### 3.1 Instruction Parser

The instruction parser converts high-level instructions into a sequence of subtasks represented by programs. Existing works [6, 5, 20, 22, 4, 28, 29] use expert trajectories with subtasks annotations as supervision because they are easy to obtain as stated in [6]. Following this strategy, we fine-tune a pre-trained BERT model [54] learned the mapping from a high-level instruction to a sequence of subtasks (*e.g.*, "put a clean tomato on the diningtable" → "(Pickup, Tomato), (Put, SinkBasin), ...") leveraging the subtasks sequences annotations in ALFRED [1]. For each subtask, the instruction parser predicts the arguments, which are the same as in [6]: (i) "obj" for the object to be picked up, (ii) "recep" for the receptacle where "obj" should be ultimately placed, (iii) "sliced" for whether *"obj"* should be sliced, and (iv) "parent" for tasks with intermediate movable receptacles (*e.g.*, "cup" in "Put a knife in a cup on the table"). After we get the subtask programs, we extract the language embeddings $e \in \mathbb{R}^{768}$ of the object words in all subprograms through a pretrained Bert model (bert-base-uncased) [55] for the follow-up concept learner module.

### 3.2 Embodied Concept Learner

Humans, even at a very early age, naturally perceive and parse the scene as objects for further understanding, *i.e.*, grouping pixels to regions without knowing their semantics [56, 57]. They then

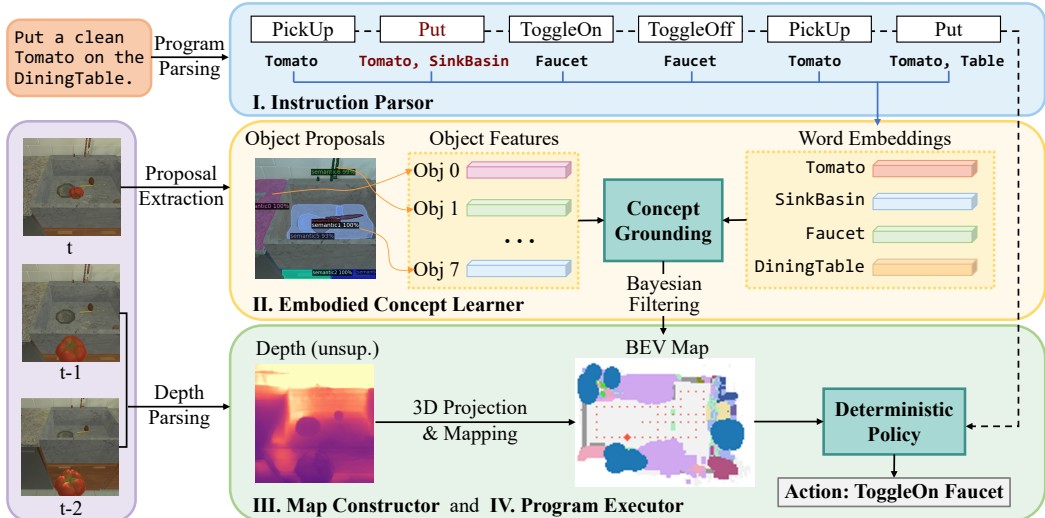

Figure 2: The framework of ECL. (i) Given a natural language goal, the instruction parser first parses it into a sequence of executable programs. (ii) The embodied concept learner extracts regional proposals in current frame and align them with the learned concepts. (iii) The map constructor then builds up semantic maps based on estimated depths and grounded visual concepts. (iv) Having the semantic maps and executable programs, the program executor predicts the agent's next action with a deterministic policy.

learn the object concepts from active interactions or expert demonstrations. Similarly, the embodied concept learner leverages an object proposal network [33] without category labels and grounds the object semantics from subgoal programs. There are two cases to be considered: 1) If a subgoal completes, the object and its corresponding receptacle objects must be displayed in the current visual frame, and most likely in adjacent frames. In this way, the concept of these objects can be grounded. For example, "go to microwave", "put the mug on the coffeemachine", and "put a mug with a pen in it on the shelf" involve 1, 2, and 3 objects, respectively. We sample visual data from four frames before completing the subtask and two frames after it to learn the visual concept based on the corresponding action descriptions. 2) If the robot agent acts "Pickup an object", the object appears in visual observation until the robot drops it. The two types of interaction data are merged and shuffled and used as input to our embodied concept learner.

Concretely, let $\{o_1, o_2, ..., o_k\}$ denotes $k$ objects detected in an visual input, and $\{f_1, f_2, ...f_k\}$ is their corresponding feature representations from the last layer of the object proposal network ($f \in \mathbb{R}^{1024}$). Let $\{e_1, e_2, ..., e_l\}$ represents $l$ word embeddings in a subgoal (program representation, $e \in \mathbb{R}^{768}$, stated in Sec. 3.1). We first project the visual representation $f$ into the semantic space $f' \in \mathbb{R}^{768}$ where the word embeddings reside by a two-layer perceptron (MLPs). The MLPs have dimensions of $1024 \rightarrow 1024 \rightarrow 768$ with Layer Normalization [58] and GELU activation [59] between the two layers. We then leverage the Hungarian maximum matching algorithm [60] for the $k$-$l$ matching, and a $\min(k, l)$ object visual representations can be matched with their word embeddings. Given an assignment matrix $x \in \mathbb{R}^{k \times l}$, the task could be formulated as a minimum cost assignment problem mathematically as follows:

$$\min_x \sum_{i=1}^{k} \sum_{j=1}^{l} d(f_i', e_j)x_{ij} \quad \text{s.t.} \quad \sum_{i=1}^{k} x_{ij} = 1, \sum_{j=1}^{l} x_{ij} \in \{0,1\}, x_{ij} \in \{0,1\}, \tag{1}$$

where $d(\cdot)$ denotes the mean square error (MSE) and we assume $l < k$ here, vice versa. In this way, we solve a min-min optimization problem by Hungarian matching, where the first minimization is used to find the best match among the two sets of features (Hungarian matching); and the second minimization is to optimize a smaller L2 loss on the matching for learning better projected representation $f'$. Both the mapping function (MLPs) and the matching are learned at the same time. Thus the MLP and the matching matrix $x$ is jointly learned from Hungarian matching.

During inference, we project each object proposal representation into the semantic space and perform nearest neighbor search (NNS) to assign a category label for it. We also calculate a soft class probability $p_i$ for the i-th object by $\mathrm{softmax}\left(\{0.1/d_{ij}\}_j\right)$, where $d_{ij}$ is the retrieval distance between the i-th object feature and the j-th word embedding. The semantic probability $\mathbf{p}$ will be used for 1) Bayesian filtering in mapping and 2) statistics of the most likely location of each type of object as a navigation policy.

### 3.3 Map Constructor

Human beings understand the semantics and layouts of space, *e.g.*, a room, mainly by first moving around, then perceiving the depth (geometry), and finally building up a semantic virtual map in our mind [61]. To mimic this process, we propose a semantic map construction module leveraging the unsupervised depth learning technique [62, 63] and probabilistic mapping inspired by Bayesian filtering. Concretely, we first train a monocular depth estimation network unsupervisedly, leveraging the photometric consistency [62] among adjacent RGB observations captured by a roaming agent. We use the unsupervised depth estimation for map construction. To build up the map, we represent the scene as voxels. Each voxel maintains a semantic probability vector $\mathbf{p}_v$ (obtained from Sec. 3.2) and a scalar variable $\sigma_v$ that represents the measurement uncertainty of this voxel. As the new depth observation come in, we first project it to 3D space as a 3D point cloud and then transform it into the map space according to the agent ego-motion. The transformed point cloud is voxelized for the follow-up map fusion. We denote the newly observed point clouds (after voxelization) as $S = \{(\mathbf{p}_s, \sigma_s)\}_{s=1}^{|S|}$ and the current voxel map as $M = \{(\mathbf{p}_m, \sigma_m)\}_{m=1}^{|M|}$. The newly observed voxels are fused to update the previous map as:

$$\mathbf{p}_m \leftarrow \frac{\sigma_s^2}{\sigma_s^2 + \sigma_m^2}\mathbf{p}_m + \frac{\sigma_m^2}{\sigma_s^2 + \sigma_m^2}\mathbf{p}_s, \ \sigma_m \leftarrow (\sigma_s^{-2} + \sigma_m^{-2})^{-\frac{1}{2}}. \tag{2}$$

Here, we assume $\mathbf{p}_s$ and $\mathbf{p}_m$ are the semantic log probability vectors (obtained from Sec. 3.2) belonging to a pair of corresponding voxels in the new frame and the current map respectively. $\sigma_s$ and $\sigma_m$ are the estimated variances of these two voxels. Initially, the variance $\sigma_s$ of the observed voxel is predicted by a CNN. This CNN is trained with the depth estimation network in an unsupervised manner by assuming a Gaussian noise model following [64]. The uncertainty-aware mapping makes it possible to correct previous mapping errors as the exploration goes on. Our probabilistic mapping is proven to be essential especially when the depth measurements are erroneous.

### 3.4 Program Executor

After concept learning and mapping, we take the average semantic probability map from demonstrations as our navigation policy. It indicates the location where each type of object most likely exists. Although the previous work FILM [6] trains a semantic policy model to predict the possible location of an object given a part of the semantic layout, the model is likely to be over-fitting. In contrast, our semantic policy is the averaged semantic map based on statistics without training, producing stable results. As shown in Fig. 2, given the predicted subprogram, the current semantic map, and a search goal sampled from the semantic policy (averaged semantic map), the deterministic policy outputs a navigation or interaction action.

The deterministic policy is defined as follows. If the object needed in the current subtask is observed in the current semantic map, the location of the object is selected as the goal; otherwise, we sample the location based on the distribution of the corresponding object class in our averaged semantic map as the goal. The robot agent then navigates towards the goal via the Fast Marching Method [65] and performs the required interaction actions.

## 4 Experiments

We show the effectiveness of each component of ECL on the ALFRED [1] benchmark. For the EIF task, we report Success Rate (SR), goal-condition success (GC), path length weighted SR (PLWSR),

Table 1: Comparison with other methods on ALFRED benchmark. The upper part contains unsupervised methods while the lower part contains the supervised counterparts with semantic or depth supervisions. We also report the ECL-Oracle model as an upper bound, with supervised segmentation and depth. The top scores are in **bold**. **Red** denotes the top success rate (SR) (ranking metric of the leaderboard) on the `test_unseen` set.

| Method | Supervision | | Test Seen | | | | Test Unseen | | | |
| --- | --- | --- | --- | --- | --- | --- | --- | --- | --- | --- |
| | Semantic | Depth | PLWGC (%) | GC (%) | PLWSR (%) | SR (%) | PLWGC (%) | GC (%) | PLWSR (%) | SR (%) |
| SEQ2SEQ [1] | × | × | 6.27 | 9.42 | 2.02 | 3.98 | 4.26 | 7.03 | 0.08 | 3.90 |
| MOCA [2] | × | × | 22.05 | 28.29 | 15.10 | 22.05 | 9.99 | 14.28 | 2.72 | 5.30 |
| LAV [3] | × | × | 13.18 | 23.21 | 6.31 | 13.35 | 10.47 | 17.27 | 3.12 | 6.38 |
| E.T. [4] | × | × | **34.93** | **45.44** | **27.78** | **38.42** | 11.46 | 18.56 | 4.10 | 8.57 |
| ECL (OURS) | × | × | 9.47 | 18.74 | 4.97 | 10.37 | **11.50** | **19.51** | **4.13** | **9.03** |
| EMBERT [18] | √ | × | **32.63** | 38.40 | 24.36 | 31.48 | 8.87 | 12.91 | 2.17 | 5.05 |
| LWIT [19] | √ | × | 23.10 | 40.53 | **43.10** | 30.92 | **16.34** | 20.91 | 5.60 | 9.42 |
| HITUT [22] | √ | × | 17.41 | 29.97 | 11.10 | 21.27 | 11.51 | 20.31 | 5.86 | 13.87 |
| ABP [20] | √ | × | 4.92 | **51.13** | 3.88 | **44.55** | 2.22 | 24.76 | 1.08 | 15.43 |
| VLNBERT [21] | √ | × | 19.48 | 33.35 | 13.88 | 24.79 | 13.18 | 22.60 | 7.66 | 16.29 |
| HLSM [5] | √ | √ | 11.53 | 35.79 | 6.69 | 25.11 | 8.45 | 27.24 | 4.34 | 16.29 |
| ECL W. DEPTH (OURS) | × | √ | 12.34 | 27.86 | 8.02 | 18.26 | 11.11 | **27.30** | 7.30 | **17.24** |
| ECL-ORACLE (OURS) | √ | √ | 15.19 | 36.40 | 10.56 | 25.90 | 13.08 | 35.02 | 9.33 | 23.68 |

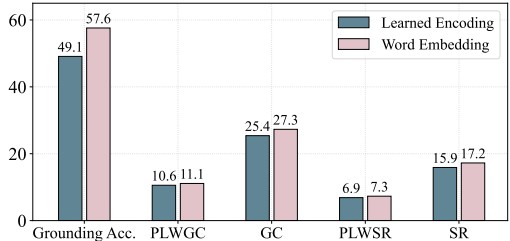

Figure 3: Results with different language representations in concept learning on `test_unseen`.

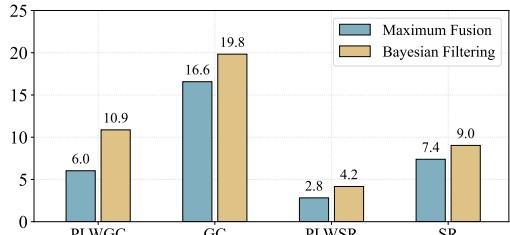

Figure 4: Evaluation with different semantic mapping techniques on `test_unseen`.

and path length weighted GC (PLWGC) as the evaluation metrics on both seen and unseen environments. SR is a binary indicator of whether all subtasks were completed. GC denotes the ratio of goal conditions completed at the end of an episode. Both SR and GC can be weighted by (path length of the expert trajectory)/(path length taken by the agent), which are called PLWSR and PLWGC. We also report the (grounding) accuracy for the concept learning and downstream reasoning tasks. More details of the benchmark and the training settings for each component can be found in Appendix.

## 4.1 Embodied Instruction Following on ALFRED

The results on ALFRED are shown in Tab. 1. ECL achieves new state-of-the-art (SR: 9.03 vs. 8.57) on the `test_unseen` set when there are no semantic and depth labels. Though counterparts [4, 2] have better performance on `test_seen`, they are likely to be over-fitting by simply memorizing the visible scenes. However, our ECL achieves stable results between the `test_seen` set and unseen set, demonstrating its generalizability. In Fig. 5, we show a trajectory to execute "place a washed sponge in a tub" and the intermediate estimates generated by ECL.

When depth supervision is used, our ECL w. depth model has a 17.24% success rate on the `test_unseen` set, as well as competitive goal-condition success rate and path length weighted results. Note that FILM [6] leverages additional dense semantic maps as supervision to train a policy network, hence not apple-to-apple comparable to our work. We report the ECL-Oracle model as an upper bound, which learns supervised segmentation and depth, and can be seen as a variant of FILM [6] without the policy network. It achieves 23.68% SR on `test_unseen`.

**Ablation Study.** We conduct experiments to study the effect of the language representation in concept learning, and the mapping strategy in map construction. The results are shown in Fig. 3 and Fig. 4, offering 1) benefiting from the natural structure of language, the word embedding is better than the learned encoding, and 2) Bayesian filtering outperforms maximum fusion as the soft probabilities could correct wrong labels.

Instruction: Place a washed sponge in a tub.

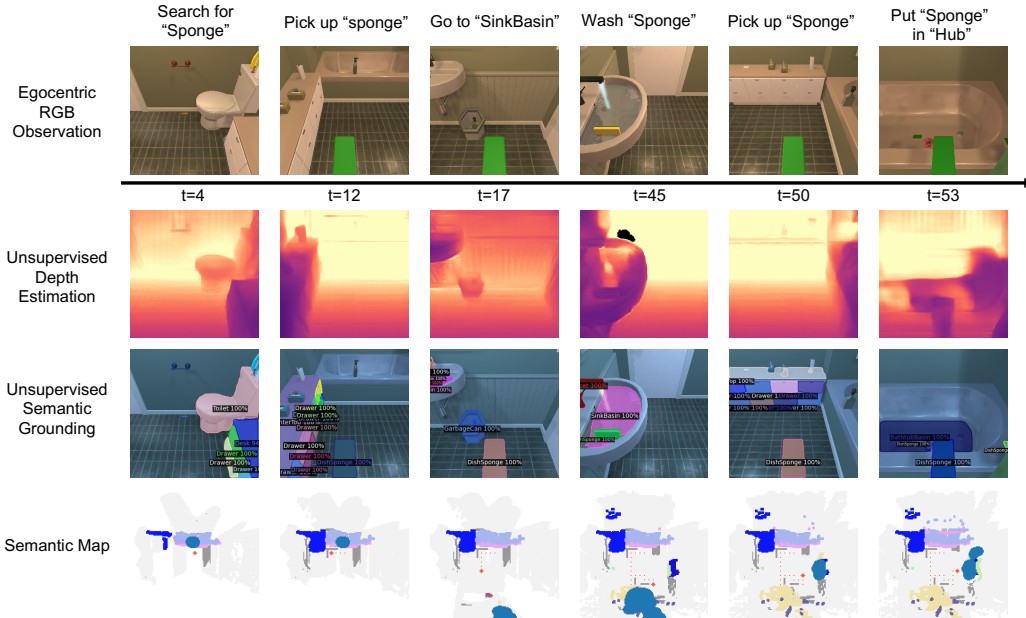

Figure 5: Visualization of intermediate estimates by ECL when an agent tries to accomplish an instruction. Based on the RGB observations, our system estimates the depths and semantic masks. The BEV semantic map is gradually established with these estimates as the exploration goes on. The goals (sub goal/final goal) are represented by big blue dots in the semantic map, while the agent trajectories are plotted as small red dots.

Table 2: The percentage of failure cases belonging to different failure modes on validation set.

| Error mode | Seen % | Unseen % |
|---|---|---|
| Grounding error/Target not found | 36.38 | 28.53 |
| Interaction failures | 6.59 | 10.39 |
| Collisions | 4.34 | 4.43 |
| Blocking/Object not accessible | 31.29 | 39.75 |
| Others | 21.41 | 16.90 |

Table 3: Concept reasoning accuracy. We leverage ECL to infer whether an object exists or to count its number in a scene.

| Model | Grounding % | Exist % | Count % |
|---|---|---|---|
| Random Guess | – | 50.0 | 25.0 |
| C3D [66] | – | 78.1 | 34.4 |
| ECL (Ours) | 57.6 | 90.6 | 56.3 |

## 4.2 Evaluation of Concept Learning

**Quantitative Evaluation.** We report the per-task evaluation results in Fig. 6. The concept learning accuracies of objects "HandTowel", "Laptop", "Bowl", and "Knife" are above 80%, because these objects frequently appear alone in the scene (easy to learn and less likely to be confused). Objects like "Basketball", "Glassbottle", "Cellphone", and "Teddybear" are rarely shown in the environment, thus their concepts are difficult to learn. We also notice that the object "apple" appears very rarely, but our model grounds its concept well with the help of language embeddings, *e.g.*, the relationship between "tomato" and "apple".

**Error Modes.** Tab. 2 shows the error mode of ECL w. depth on ALFRED validation set. We see that "blocking and object not accessible" is the most common error mode, which is mainly caused by incorrectly estimated depth or undetected visual objects/concepts. Additionally, around 30% of the failures are due to wrongly grounded concepts or the target object not being found. If we replace our unsupervised concept learning with supervised semantics (ECL-Oracle), the percentage of the error mode for "Grounding error/Target not found" changes to 7.38% and "blocking and object not accessible" becomes 44.00%.

**Visualization.** We visualize our concept learning results in Fig. 8 by showing the original image, the supervised learned semantics, and our grounded semantics by the concept learner. We observe our concept learning keeps more object proposals than the supervised model. While most of the

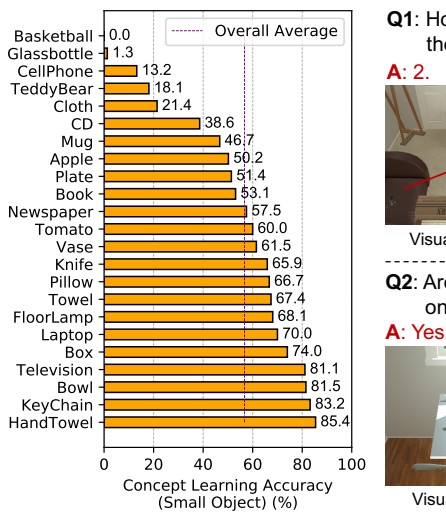

Figure 6: Concept learning accuracy. Results for challenging small objects are shown. Complete analyses are in appendix.

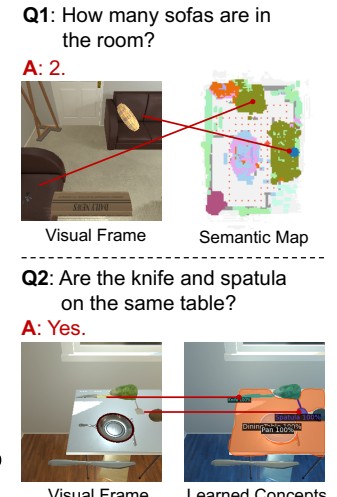

**Q1**: How many sofas are in the room?
**A**: 2.

Visual Frame          Semantic Map

**Q2**: Are the knife and spatula on the same table?
**A**: Yes.

Visual Frame          Learned Concepts

Figure 7: Examples of concept reasoning by ECL: the count task and the high-level question-answering.

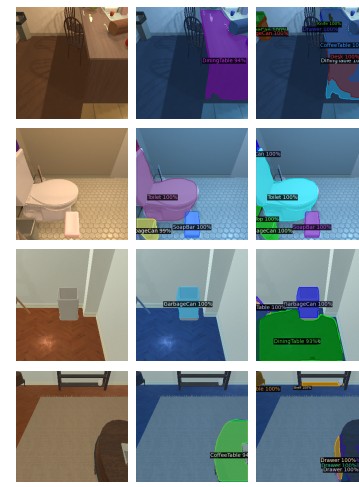

Figure 8: Concept learning visualization. From left to right: the original image, supervised instance segmentation map, and our concept learning results.

main objects in an image can be grounded correctly, there exist a few wrong labels in overlapped or corner areas. We also show two failure cases on the third and fourth rows of Fig. 8. The first one recognizes "floor" as "diningtable", a bug that could be fixed by our Bayesian filtering-based semantic mapping. The other one identifies "coffeetable" as "drawer", which causes the error "target not found". The instruction would succeed if we take the ground truth concept for "coffeetable".

### 4.3 Concept Reasoning

In addition to EIF, we show the learned concept can be transferred to embodied reasoning tasks, *e.g.*, (i) the existence of objects in the scene, (ii) count the number of objects in the scene (Fig. 7). We build the reasoning dataset by randomly sampling 16 objects from 10 scenes, of which 8 scenes are used for training and the other 2 for testing. A naïve baseline is random guessing with 50% accuracy for the exist task and 25% accuracy for the count task. We also train a C3D model [66] that samples 16 frames as input and outputs predictions directly. Our ECL performs clear and step-by-step interpretable reasoning through semantic grounding and mapping. As Tab. 3 shows, it outperforms both baselines by a large margin. By embodied concept learning, ECL can also resolve high-level 3D question-answering tasks, like "whether two objects appear on a table" in Fig. 7.

## 5 Discussions

This paper proposes ECL, a general framework that can ground visual concepts, build semantic maps and plan actions to accomplish tasks by learning purely from human demonstrations and language instructions. While achieving good performance on embodied instruction following, ECL has limitations. Although the ALFRED benchmark is photo-realistic, comprehensive, and challenging, there still exists a gap between the embodied environment and the real world. We leave the physical deployment of the framework as our future work.

**Acknowledgements.** This work is supported by MIT-IBM Watson AI Lab and its member company Nexplore, Amazon Research Award, ONR MURI, DARPA Machine Common Sense program, ONR (N00014-18-1-2847), and Mitsubishi Electric. Ping Luo is supported by the General Research Fund of HK No.27208720, No.17212120, and No.17200622.

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
