# OpenReview forum: "Embodied Concept Learner: Self-supervised Learning of Concepts and Mapping through Instruction Following"
_robot-learning.org/CoRL/2022/Conference — CoRL 2022 Poster_

### Official Review · Reviewer_TjRQ · 2022-07-25

**Originality:** Excellent
**Technical Quality:** Very Good
**Clarity Of Presentation:** Excellent
**Impact:** 4

**Recommendation:**

Strong Accept: I recommend accepting the paper and will argue for my recommendation even if other reviewers hold a different opinion.

**Summary:**

This paper addresses the problem of embodied concept learning, whose aim is to learn concepts through interaction in an interactive 3D environment. Specifically, this work focuses on learning to ground visual concepts from human demonstrations and language instructions and enabling embodied instruction following. To this end, the paper proposes a framework that parses an instruction into a sequence of subtasks, grounds visual concepts based on language descriptions, leverages learned concepts to estimate the depth and construct semantic maps, and fulfills each subtask with a crafted rule-based policy. The experiments on the ALFRED benchmark show that the proposed framework outperforms baselines when no depth and semantic label is provided in terms of success rate and goal-condition success. I believe this work explores a promising research direction (i.e. embodied concept learning) and presents an effective framework to address the problem.

**Issues:**

Described in the Strengths And Weaknesses section.


**Quality Of The Limitations Section:**

Additional details required

**Reviewer Expertise:**

4: The reviewer is confident but not absolutely certain that the evaluation is correct

**Robotics Focus:**

Highly relevant to robotics but no hardware experiments

**Strengths And Weaknesses:**

## Paper strengths and contributions
**Motivation and intuition**
- The motivation for studying embodied concept learning is convincing.
- Studying embodied concept learning from and for embodied instruction following task seems like a decent choice.

**Technical contribution**
- Utilizing an objects proposal network to obtain detected objects makes sense and seems to be effective
- Using the Hungarian maximum matching algorithm seems effective for matching detected objects and extracted word embeddings.
- The map constructor that builds up semantic maps based on estimated depths and grounded visual concepts seems to incorporate the information well.

**Clarity**
The overall writing is very clear. The authors utilize figures and examples well to illustrate the ideas.

**Experimental results**
- The presentation of the experimental results is very clear.
- The experimental results on ALFRED show that the proposed framework (1) outperforms baselines when no depth or semantic label is available, (2) generalizes better between test-seen and test-unseen, and (3) achieves competitive performance when depth supervision is used.
- The qualitative results in both the main paper and the appendix are well-analyzed and interesting.

**Ablation study**
Ablation studies justify the effect of the language representation in concept learning and the mapping strategy in map construction.

## Paper weaknesses and questions

**Oversell**
I feel the authors oversell the paper a little bit. After reading the abstract and the introduction,
I thought the proposed framework learns from scratch from only interactions and human demonstrations.
Yet, word embeddings from a pre-trained BERT model are utilized which can provide relationships among objects and an object proposal network is leveraged which has learned low-level visual features and how to group pixels.
In my opinion, the authors can tone down the key motivation a little bit so that the readers do not get disappointed.

**Instruction-program pairs**
This work fine-tunes a pre-trained BERT model to convert an instruction to a program (i.e. a sequence of subtasks) using paired data from the subtask sequence annotations in ALFRED. I wonder if the instruction parser can learn from the task completion signal when there is no instruction-program pair available, or if paired data is just required.

**Instruction parser's accuracy**
I wonder how accurate the instruction parser is. It would be great if this information can be provided.

**Training object proposal network**
How was the object proposal network trained? Was it trained on data from ALFRED? I wonder how accurate it has to be to allow the whole framework to work. Also, is it fine-tuned during the learning?

**Learning continuous concepts**
This work presents an effective framework to ground visual concepts by matching them to language descriptions in a "discrete" manner. I wonder how we can learn continuous concepts or even dynamic concepts.

**Deterministic (rule-based) policy**
This work incorporates a simple deterministic policy. I wonder how this can scale up to environments that require longer-term exploration.

**Related work**
I feel the authors can include the following papers to make the related work more comprehensive:
- Synthesizing Environment-Aware Activities via Activity Sketches: it generates program sketches from instructions.
- Learning to Synthesize Programs as Interpretable and Generalizable Policies: it discusses interpretability and generalizability.

====== After rebuttal ======

I thank the authors for the response and the revision, which addresses of my concerns. I have carefully read other reviews and decided to keep my original rating.



**Summary Of Recommendation:**

I believe this work explores a promising research direction (i.e. embodied concept learning) and presents an effective framework to address the problem. While I have a few minor concerns stated above, I enjoy reading this paper and would love to see this paper included at the conference.

---

> ### Author Response · Authors · 2022-08-27
> **Response to Reviewer #TjRQ -- Part 2**
>
> **Q5. [Learning continuous concepts]** This work presents an effective framework to ground visual concepts by matching them to language descriptions in a "discrete" manner. I wonder how we can learn continuous concepts or even dynamic concepts. \
> **A:** Good suggestion. We take the soft matching score in this work when constructing the map by Bayesian filtering. Further employing the soft scores or continuous concepts in the program executor could be a good choice. We will take it as future work to learn dynamic concepts, following [C]. We can extract the video-based temporal feature and then align them to dynamic concepts.
>
> [C] Chen Z, Mao J, Wu J, Wong KY, Tenenbaum JB, Gan C, Hassanzadeh O, Bhattacharjya D. Grounding physical object and event concepts through dynamic visual reasoning. In International Conference on Learning Representations 2021.
>
>
>
> **Q6. [Deterministic (rule-based) policy]** This work incorporates a simple deterministic policy. I wonder how this can scale up to environments that require longer-term exploration. \
> **A:** Our deterministic policy is based on the Fast Marching Method [56], which is well-studied and we believe that it can be extended to other scenarios. For longer-term exploration, the bottleneck for executors may lie in finding the proper goal location. We can train another policy network that, given the current partial observation, it predicts where the goal location is likely to be.
>
>
> **Q7. [Related work]** I feel the authors can include the following papers to make the related work more comprehensive: (1) Synthesizing Environment-Aware Activities via Activity Sketches: it generates program sketches from instructions. (2) Learning to Synthesize Programs as Interpretable and Generalizable Policies: it discusses interpretability and generalizability. \
> **A:** Thanks for letting us know the good papers. We have carefully studied the papers and added them in our paper.
>
>
> \
> An updated manuscript is attached. Thanks again for your time and effort! For any other questions, please feel free to let us know during the rebuttal window.
>
> Best, \
> Authors

---

> ### Author Response · Authors · 2022-08-27
> **Response to Reviewer #TjRQ -- Part 1**
>
> Dear Reviewer,
>
> Thank you for the positive comments and insightful suggestions.
>
> **Q1. [Oversell]** I feel the authors oversell the paper a little bit. After reading the abstract and the introduction, I thought the proposed framework learns from scratch from only interactions and human demonstrations. Yet, word embeddings from a pre-trained BERT model are utilized which can provide relationships among objects and an object proposal network is leveraged which has learned low-level visual features and how to group pixels. In my opinion, the authors can tone down the key motivation a little bit so that the readers do not get disappointed. \
> **A:** Thanks for the suggestion. We have toned down our claims.
>
> In addition to expert demonstrations, we use two pretrained models (an object proposal network without semantics and a language BERT model pretrained on Wikipedia datasets, which can be seen as the most basic language and perceptual abilities of babies) in our framework but without using semantic or depth labels in ALFRED. The discussions have been delivered in our paper to avoid overclaiming and make our paper more readable. Thank you again!
>
>
> **Q2. [Instruction-program pairs]** This work fine-tunes a pre-trained BERT model to convert an instruction to a program (i.e. a sequence of subtasks) using paired data from the subtask sequence annotations in ALFRED. I wonder if the instruction parser can learn from the task completion signal when there is no instruction-program pair available, or if paired data is just required. \
> **A:** Good point! In this work, we use paired data to train the BERT model. Directly learning instruction parser from the task completion signal could be a better choice to reduce supervision. We have tried some unsupervised decomposition methods [A, B] but it's still difficult in ALFRED by only high-level instructions. We will take this interesting direction as our future work.
>
> [A] Lu Y, Shen Y, Zhou S, Courville A, Tenenbaum JB, Gan C. Learning task decomposition with ordered memory policy network. arXiv preprint arXiv:2103.10972. 2021 Mar 19. \
> [B] Luo Z, Mao J, Wu J, Perez T, Tenenbaum JB, Kaelbling LP. Learning Rational Skills for Planning from Demonstrations and Instructions.
>
>
> **Q3. [Instruction parser's accuracy]** I wonder how accurate the instruction parser is. It would be great if this information can be provided. \
> **A:** Learning programs from the instruction is a well-studied NLP problem. The accuracy of our instruction parser is 95.3%.
>
>
> **Q4. [Training object proposal network]** How was the object proposal network trained? Was it trained on data from ALFRED? I wonder how accurate it has to be to allow the whole framework to work. Also, is it fine-tuned during the learning? \
> **A:** We load a pretrained object proposal network training on the COCO dataset, and then finetune it using the object proposals in ALFRED without any semantics. In this work, we use as many proposals as possible and smaller thresholds to include all objects as possible, but leads to duplicate objects, see Figures 1 and 5. The object proposal network is then fixed and not fine-tuned during the learning.

---

### Official Review · Reviewer_XMVt · 2022-07-27

**Originality:** Very Good
**Technical Quality:** Very Good
**Clarity Of Presentation:** Good
**Impact:** 3

**Recommendation:**

Weak Accept: I recommend accepting the paper, but will not argue for my recommendation if the majority of other reviewers have a different opinion.

**Summary:**

This paper presents Embodied Concept Learner (ECL), a self-supervised framework for grounding concepts and following instructions in 3D environments. The framework is set in the ALFRED instruction following environment. Instead of relying on ground-truth segmentations, ECL makes connections between instructions, observations, and actions to learn object-centric concepts like “knife” and “soap bottle”. Similarly, for estimating depth, the method uses an unsupervised-learning approach. Together, the learnt concepts and depth predictions are fused with a Bayes filter to build a spatial-semantic map of the scene. This map is used with an executor module to solve instruction following tasks. The experiments include evaluations on the ALFRED benchmark with metrics like success and path-weighed success on seen and unseen scenes. While the proposed approach does not achieve state-of-the-art performance, it achieves compelling results without using any ground-truth segmentations or depth.

**Issues:**

See above on the concern regarding the approach being specific to ALFRED and missing details.

**Quality Of The Limitations Section:**

Limitations are addressed clearly

**Reviewer Expertise:**

4: The reviewer is confident but not absolutely certain that the evaluation is correct

**Robotics Focus:**

Relevant but unlikely to deploy to hardware in near future

**Strengths And Weaknesses:**

**Strengths**

+ Overall, the idea of using self-supervised learning to learn object-centric concepts to aid embodied agents is an interesting and novel approach. Most existing methods for the ALFRED challenge make heavy use of ground-truth segmentation masks and perfect depthmaps from the simulator. Such supervision is often hard to obtain in real-world robotics where it’s difficult to pre-define a set of object categories, or obtain noiseless depth estimates from onboard sensors. As such, ECL takes a good step towards realizing approaches that have been prevalent in the Embodied AI community for instruction following tasks.

+ The experiments include a good set of comparisons to prior work on the ALFRED benchmark. Additionally, the results also include ablations investigating various aspects of ECL like Bayesian Filtering, word embeddings, and an analysis on common failure cases. If the authors open-source their implementation, these results should be easily reproducible.

+ The results in Section 4.2 and 4.3 are interesting. Without any explicit object-level supervision, the learnt concepts can be re-used to query questions like “how many sofas are in the room?”. Future works could follow up these investigations by studying the generalizability of bottom-up perceptual representations that are learnt not through explicit supervision, but from environmental interactions.

**Weaknesses**

- One major concern is that ECL’s approach to self-supervised concept learning might be a bit specific to the ALFRED environment. As described in Section 3.2, the training data is composed of frames from before and after interacting with objects. These interactions are observed through discrete changes in image observations, e.g. picking up an object immediately zaps the object to the agent’s visual frame in the center. Such obvious visual cues might not be apparent in real-world settings, especially if the agent is interacting with objects through a manipulator.

- Section 3.2 is missing details on how exactly the concept learner is trained. The word embeddings and visual representations are fine-tuned with MLP layers, but they are matched with the Hungarian algorithm. How is gradient descent used with the Hungarian algorithm? And what is the training loss? Without these details, it’s hard to understand how exactly ECL works in practice. Similarly, is the Program Executor (in Section 3.4) a hand-coded policy specifically designed for ALFRED tasks? If there is no learning or planning involved here, again, such hand-coded policies might be specific to ALFRED-only tasks.

- One minor concern regarding the prose is the use of frequent analogies comparing ECL’s approach to “human-like” and “baby-like” learning. While it’s great that ECL is inspired by theories in cognitive science, and a lot of its core ideas are rooted in cognitive development, some references like “Humans … group[ing] pixels regions without knowing their semantics” and making an “agent behave like a baby” with a Bayes filter for mapping – is unsubstantiated. These analogies could be rephrased or included with proper citations to cognitive science literature to back-up the claims.


**Summary Of Recommendation:**

ECL presents a novel approach for learning object-centric concepts without explicit segmentation masks or depth supervision. ECL’s contributions might be limited to ALFRED tasks, but nonetheless they might be broadly useful to the embodied AI and robotics community.

**Post Rebuttal**
These responses address most of my concerns. I am still a bit skeptical about some of the cogsci claims made in the paper like "humans ... [group]ing pixels" and equating Bayes-filter mapping to how humans map and navigate. I checked out the references the authors mentioned, and I couldn't find any evidence of "pixel" representations or Bayesian-filtering. But overall, I think the contributions here are still of interest to the embodied-AI/robotics community, and so I will keep my score of Weak Accept.

---

> ### Author Response · Authors · 2022-08-27
> **Response to Reviewer #XMVt -- Part 2**
>
>
>
> **Q3. [One minor concern regarding the prose is the use of frequent analogies comparing ECL’s approach to “human-like” and “baby-like” learning. While it’s great that ECL is inspired by theories in cognitive science, and a lot of its core ideas are rooted in cognitive development, some references like “Humans … group[ing] pixels regions without knowing their semantics” and making an “agent behave like a baby” with a Bayes filter for mapping – is unsubstantiated. These analogies could be rephrased or included with proper citations to cognitive science literature to back-up the claims.]** \
> **A:** Thank you for the suggestion! We have added papers [A, B, C] from cognitive science to back up the claims in our paper.
>
>
> [A] Maguire EA, Burgess N, Donnett JG, Frackowiak RS, Frith CD, O'Keefe J. Knowing where and getting there: a human navigation network. Science. 1998 May 8;280(5365):921-4. \
> [B] Regan D. Human perception of objects. Sunderland, MA: Sinauer; 2000. \
> [C] Carbon CC. Understanding human perception by human-made illusions. Frontiers in human neuroscience. 2014 Jul 31;8:566.
>
>
> \
> An updated manuscript is attached. Thanks again for your time and effort! For any other questions, please feel free to let us know during the rebuttal window.
>
> Best, \
> Authors

---

> ### Author Response · Authors · 2022-08-27
> **Response to Reviewer #XMVt -- Part 1**
>
> Thank you for the positive comments and insightful suggestions.
>
> **Q1. [One major concern is that ECL’s approach to self-supervised concept learning might be a bit specific to the ALFRED environment. As described in Section 3.2, the training data is composed of frames from before and after interacting with objects. These interactions are observed through discrete changes in image observations, e.g. picking up an object immediately zaps the object to the agent’s visual frame in the center. Such obvious visual cues might not be apparent in real-world settings, especially if the agent is interacting with objects through a manipulator.]** \
> **A:** Thank you for pointing it out. We agree with you that we have some assumptions based on the artifact of the AI2THOR environment. The assumptions include:
> 1) If a subgoal completes, the object and its corresponding receptacle objects must be displayed in the current visual frame, and most likely in adjacent frames.
> 2) If the robot agent acts "Pickup an object", the object appears in visual observation until the robot drops it.
>
> However, the assumptions do not affect the concept learning performance and could be applied to real environments. The only requirement of our concept learner is that the object is not fully occluded and can be detected as an object proposal. It is reasonable because humans also use similar steps and observations to solve tasks (also, when a robot picks up an object, we could make the object appear in visual observation). In this way, our concept grounding does not depend on the object's location, which can be learned when an object is not in the center. For example, we can also learn the concept of receptible objects, such as tables and shelves, which do not appear in the center of the observation.
>
> The discontinuous motion problem is a limitation of ALFRED that the action execution is not feasible, i.e, picking up an object by only one command without robot manipulation. But it doesn't affect our concept learning accuracy. Moreover, the low-level control task and the current embodied instruction following task are orthogonal, which means the two tasks can still be decoupled in real-world scenarios, while our model focus on instruction following.
>
>
>
>
> **Q2. [Section 3.2 is missing details on how exactly the concept learner is trained. The word embeddings and visual representations are fine-tuned with MLP layers, but they are matched with the Hungarian algorithm. How is gradient descent used with the Hungarian algorithm? And what is the training loss? Without these details, it’s hard to understand how exactly ECL works in practice. Similarly, is the Program Executor (in Section 3.4) a hand-coded policy specifically designed for ALFRED tasks? If there is no learning or planning involved here, again, such hand-coded policies might be specific to ALFRED-only tasks.]** \
> **A:** Good question. The MLP layer is jointly learned in our concept learner by Hungarian matching. In this work, the visual feature is obtained from a pretrained object proposal network, while the work embedding is obtained from a language model. Both the two pretrained representations are fixed thus only the MLP contains learnable parameters. The MLP aims to find a proper projection to align the two data distributions (vision and language).
>
> By Hungarian matching, we solve a min-min optimization problem, where the first minimization is used to find the best match among the two sets of features (Hungarian matching); and the second minimization is to optimize a smaller L2 loss  $L = \||f'\_i - e\_j\||\_2^2$ on the matching for learning better projected representation $f'$, as shown in below (Line 151-156 of our paper):
> $\min\_x\sum\_{i=1}^k\sum\_{j=1}^l d(f'\_i, e\_j) x\_{ij}~~~\text{s.t.}~~~\sum\_{i=1}^k x\_{ij}=1, \sum\_{j=1}^l x\_{ij}\in \{0,1\}, x\_{ij} \in \{0,1\}$. \
> Both the mapping function (MLPs) and the matching are learned at the same time. Thus the MLP is jointly learned from Hungarian matching in a differentiable manner.
>
> The deterministic policy is based on the Fast Marching Method [56]. If the object needed in the current subtask is observed in the current semantic map, the location of the object is selected as the goal; otherwise, we sample the location based on the distribution of the corresponding object class in our averaged semantic map as the goal. In both cases, we did not use any domain knowledge about ALFRED. We find the goal from the concept learner and plan the shortest path to the goal based on our semantic map. It's a very general solution that can be used in many other tasks or environments rather than a hand-coded policy for ALFRED.

---

### Official Review · Reviewer_dHJH · 2022-07-28

**Originality:** Good
**Technical Quality:** Good
**Clarity Of Presentation:** Fair
**Impact:** 2

**Recommendation:**

Weak Accept: I recommend accepting the paper, but will not argue for my recommendation if the majority of other reviewers have a different opinion.

**Summary:**

This paper presents embodied concept learner that can ground the objects included in the instructions and constructs a semantic map. Grounding is achieved by a minimum cost assignment problem between visual features and word embeddings. The semantic map is constructed by assigning the object probabilities to 3D space. Based on the semantic map and parsed instructions, the agent can predict actions.

**Issues:**

Described in the strengths and weaknesses section.

**Quality Of The Limitations Section:**

Additional details required

**Reviewer Expertise:**

3: The reviewer is fairly confident that the evaluation is correct

**Robotics Focus:**

Highly relevant to robotics but no hardware experiments

**Strengths And Weaknesses:**

# Strengths
- ECL achieved high performance in unseen test tasks of ALFRED. This result indicates that the ECL has high generalizability.

# Weaknesses and suggestions
- First of all, I have a question about the purpose of this study written in lines 47-56. This study aims to learn knowledge from expert demonstrations without supervision. However, this study uses the labels in ALFRD, and this is considered different from learning by a baby.
- For a similar reason, if demonstrations mean the labels in ALFRED in this study, I think these are not demonstrations but dis ambiguated labels.
- Sec 3.2: How were the MLP that projects visual representation into semantic space of word learned? If this projection is already learned, this leads that the agent already has knowledge about the objects and their names.
-  Figure 3: I do not understand the difference between learned encoding and word embedding.
- Lines 229-231: typos?


**Summary Of Recommendation:**

I think this paper should be revised to be accepted.

---

> ### Author Response · Authors · 2022-08-27
> **Response to Reviewer #dHJH**
>
> Dear Reviewer,
>
> Thank you for the constructive comments and suggestions.
>
> **Q1. [First of all, I have a question about the purpose of this study written in lines 47-56. This study aims to learn knowledge from expert demonstrations without supervision. However, this study uses the labels in ALFRD, and this is considered different from learning by a baby.]** \
> **A:** Sorry for the confusion. In this work, we learn the language parser, object concept, and depth information, as well as the semantic probability map from expert demonstrations. Regarding using labels in ALFRED, we make detailed clarifications here to make sure we are on the same page.
>
> - The instruction parser is learned from language goals in demonstrations, which is the only supervisedly learned part. And previous works [5,6,21] train the program parser in the same way.
> - The embodied concept learner is unsupervisedly learned by matching the visual representations with language tokens. Here we use two general pretrained models, an object proposal network without semantics and a language BERT model pretrained on Wikipedia datasets. The two general pretrained models work similarly to the most basic language and perceptual abilities of babies, without the need to learn on ALFRED.
> - The map constructor and program executor are learned unsupervisedly by leveraging the output of the concept learner and temporal consistency between two consecutive frames.
>
> In summary, in addition to expert demonstrations, we use two pretrained models in our framework but without using semantic or depth labels in ALFRED. The discussions have been delivered in our paper to avoid overclaiming and make our paper more readable. Thanks.
>
>
> **Q2. [For a similar reason, if demonstrations mean the labels in ALFRED in this study, I think these are not demonstrations but dis ambiguated labels.]** \
> **A:** Thanks. In addition to expert demonstrations (actions and language instruction), we use two pretrained models in our framework, an object proposal network and a language word embedding model, which can be seen as the most basic language and perceptual abilities of babies.  We claimed our method as self-supervised because our framework does not need semantic and depth labels for training; the goal is to distill the semantic and geometry knowledge from the given demonstrations. We have toned down our claim and made it clearer in our updated manuscript (see attached).
>
>
> **Q3. [Sec 3.2: How were the MLP that projects visual representation into semantic space of word learned? If this projection is already learned, this leads that the agent already has knowledge about the objects and their names.]** \
> **A:** Thanks. The projection is learned in our concept learner by Hungarian matching. In this work, the visual feature is obtained from a pretrained object proposal network, while the word embedding is obtained from a language model. The MLP aims to find a proper projection to align the two data distributions (vision and language). In this way, the agent does not have knowledge about the matching of objects and their names.
>
> By Hungarian matching, we solve a min-min optimization problem, where the first minimization is used to find the best match among the two sets of features; and the second minimization is to optimize a smaller L2 loss on the matching for learning better projected representation $f'$. Thus the MLP is jointly learned from Hungarian matching in a differentiable manner.
>
>
> **Q4. [Figure 3: I do not understand the difference between learned encoding and word embedding.]** \
> **A:** "Word embedding" in Fig. 3 shows the results when we use the word embeddings (fixed during training) from the pretrained BERT language model.  For the "Learned encoding", the word embeddings are directly learned on our dataset without the BERT initializations. We found that the configuration of "word embedding" leads to better results and thus we use it by default in our experiments.
>
>
> **Q5. [Lines 229-231: typos?]** \
> **A:** Thanks for pointing it out. We have fixed the typo and updated our manuscript.
>
> \
> An updated manuscript is attached. We wish that our response has addressed your concerns, and turns your assessment to the positive side. If you have any questions, please feel free to let us know during the rebuttal window. We appreciate your suggestions and comments! Thank you!

---

### Official Review · Reviewer_Zyvv · 2022-08-01

**Originality:** Good
**Technical Quality:** Very Good
**Clarity Of Presentation:** Good
**Impact:** 3

**Recommendation:**

Weak Accept: I recommend accepting the paper, but will not argue for my recommendation if the majority of other reviewers have a different opinion.

**Summary:**

The paper proposed a system which learns depth in an unsupervised manner and semantic segmentation in a self-supervised manner from object segmentations from Mask R-CNN and task demonstrates and language instructions. Using the resulting predicted depth and semantic segmentations, the approach builds an uncertainty-aware semantic 3D map which can continually update information from new views. This map can be used for downstream task solving or question answering.

**Issues:**

 - A key limitation of this work are their core assumptions, which is that "1) If a subgoal completes, ... until the robot drops it" (L137-143). This is an artifact of the AI2THOR environment, and may not generally be applicable in other simulators and in the real world.
- From section 3.2, it is not clear how the self-supervised semantic model is trained. For instance, after reading the paper, I still have questions about why hungarian matching over a randomly initiallized MLP might make sense during the beginning of training, and how is the MLP's supervision signal derived in a self-supervised manner. Please provide more details.
- When the robot is deployed in home level tasks, achieving high enough resolution using voxel representations may cause prohibitive memory usage. While this 3D representation makes it straight forward to aggregate semantic and spatial uncertainties on a grid, it is also not scalable.
- Does it make more sense to allow ECL-Oracle access to ground truth segmentation and depth rather than to ground truth segmentation and depth supervision? This would help gauge ECL's performance if its self-supervised depth and semantic segmentation was perfect.
- Mistake in L231 about objects that appear infrequently in the environment.
- I'm curious to see exposure frequency v.s. concept learning accuracy of object classes to back up statements in L229 - 232.

**Quality Of The Limitations Section:**

Additional details required

**Reviewer Expertise:**

4: The reviewer is confident but not absolutely certain that the evaluation is correct

**Robotics Focus:**

Highly relevant to robotics but no hardware experiments

**Strengths And Weaknesses:**

Strengths:
 - Interesting approach to fusing semantic and spatial uncertainty into one unified 3D scene representation for planning.
 - Impressive quantitative results on Alfred benchmark.

Weaknesses:
- Key assumption for self-supervised semantic segmentation matching may not apply on other simulators or real robots. This assumption is also not addressed in the limitations section.
- Some training details unclear from writing.

**Summary Of Recommendation:**

Overall an interesting system which combines depth and semantic segmentation self-supervised learning with 3D map building for task solving. While their quantitative results demonstrate impressive performance on the ALFRED benchmark, I'm doubtful whether their key assumption allows the system to work on real robots.

---

> ### Author Response · Authors · 2022-08-27
> **Response to Reviewer #Zyvv -- Part 2**
>
> **Q3. [From section 3.2, it is not clear how the self-supervised semantic model is trained. For instance, after reading the paper, I still have questions about why hungarian matching over a randomly initiallized MLP might make sense during the beginning of training, and how is the MLP's supervision signal derived in a self-supervised manner. Please provide more details.]** \
> **A:** Thanks. In this work, the visual feature is obtained from a pretrained object proposal network, while the word embedding is obtained from a language model. The MLP aims to find a proper projection to match the two data distributions (vision and language).
>
> By Hungarian matching, we solve a min-min optimization problem, where the first minimization is used to find the best match among the two sets of features; and the second minimization is to optimize a smaller L2 loss on the matching for learning better projected representation $f'$. Thus the MLP is learned from Hungarian matching in a differentiable manner.
>
>
> **Q4. [When the robot is deployed in home level tasks, achieving high enough resolution using voxel representations may cause prohibitive memory usage. While this 3D representation makes it straight forward to aggregate semantic and spatial uncertainties on a grid, it is also not scalable.]** \
> **A:** Good Question! In this work, we "voxelize" the scene directly as the size of the room is relatively small. For future high-resolution scenarios, one possible improvement is to leverage the sparse voxel representation, since objects do not exist in most areas of a 3D room. Techniques like sparse voxel representation and sparse convolution have already been studied well, like [A, B], and could be directly integrated into our work.
>
> [A] Laine S, Karras T. Efficient sparse voxel octrees. IEEE Transactions on Visualization and Computer Graphics. 2010 Nov 9;17(8):1048-59. \
> [B] Liu B, Wang M, Foroosh H, Tappen M, Pensky M. Sparse convolutional neural networks. InProceedings of the IEEE conference on computer vision and pattern recognition 2015 (pp. 806-814).
>
>
> **Q5. [Does it make more sense to allow ECL-Oracle access to ground truth segmentation and depth rather than to ground truth segmentation and depth supervision? This would help gauge ECL's performance if its self-supervised depth and semantic segmentation was perfect.]** \
> **A:** This is a good point. We replace the ECL-Oracle in the paper by 1) using ground truth depth; and 2) using both ground truth segmentation and depth, and report the test unseen results as follows.
>
> |                     | PLWGC | GC    | PLWSR | SR    |
> |---------------------|-------|-------|-------|-------|
> | ECL-ORACLE          | 13.08 | 35.02 | 9.33  | 23.68 |
> | + Depth GT          | 15.77 | 40.01 | 11.75 | 28.25 |
> | + Depth GT + Seg GT | 35.25 | 57.60 | 26.82 | 43.03 |
>
> From the Table we can see that, if the self-supervised depth and semantic segmentation are provided, the performance of EC-ORACLE could be further improved. The ground truth semantics contribute more than ground truth depth to the final model, demonstrating that semantics is critical to the embodied instruction following task.
>
>
>
> **Q6. [Mistake in L231 about objects that appear infrequently in the environment.]** \
> **A:** Thanks for pointing it out. We have fixed the typo and updated our manuscript.
>
>
> **Q7. [I'm curious to see exposure frequency v.s. concept learning accuracy of object classes to back up statements in L229 - 232.]** \
> **A:** We report the exposure frequency v.s. concept learning accuracy of object classes. The figure for the comparison is attached. The Spearman correlation coefficient is 0.65, indicating that they are positively correlated. The figure of the comparison is attached. We can see that the two ranks are highly related, though some outliers exist. For example, the "knife" category appears most frequently, but the grounding accuracy is not high, because it always appears together with other categories like "fork", leading to confusion in the concept learner.
>
>
> \
> Thanks again for your time and effort! For any other questions, please feel free to let us know during the rebuttal window.
>
> Best, \
> Authors

---

> ### Author Response · Authors · 2022-08-27
> **Response to Reviewer #Zyvv -- Part 1**
>
> Dear Reviewer,
>
> Thank you for the positive comments and insightful suggestions. An updated manuscript is attached.
>
> **Q1. [Key assumption for self-supervised semantic segmentation matching may not apply on other simulators or real robots. This assumption is also not addressed in the limitations section.]** \
> **A:** Good question! Real robotics applications have been one of the longstanding motivations for this work and have been carefully considered by the authors in the design of ALFRED. When generalizing our model to real-world scenarios:
> - The instruction parser is supervisedly trained, and can be directly employed in the real world.
> - The embodied concept learner should work when there are real-world demonstrations. Currently, we have some assumptions based on the artifact of the AI2THOR environment. However, the assumptions do not affect the concept learning performance and could be applied to real environments. (The only requirement of our concept learner is that the object is not fully occluded and could be detected as an object proposal. In this way, our concept grounding does not depend on specific environments or the object's locations.)
> - Unsupervised depth and mapping are well-studied problems in the real world. We see this as a reasonable assumption for the time being.
> - Still, there are some limitations in ALFRED that the action execution is not feasible, i.e, picking up an object by only one command without robot manipulation. However, the low-level control task and the current embodied instruction following task are orthogonal, which means the two tasks can still be decoupled in real-world scenarios, while our model focus on instruction following.
>
> However, it's really challenging for a robot to perform instruction following in an unseen real-world environment, even in a simulated environment (test unseen success rate of only 23.6% even in our oracle model). To this end, ALFRED simplifies the hard problem of making meaningful progress through tight integration between visual perception, language instruction, and robotic navigation and manipulation. To the best of our knowledge, no other benchmarks contain language instructions in an interactive 3D environment with visual observation and navigation. Also, the prior works [5,6,20,21] towards this goal evaluate only on the ALFRED benchmark. As the field progresses, we are confident more works and benchmarks will be introduced, and we will take it as our future research direction.
>
>
> **Q2. [A key limitation of this work are their core assumptions, which is that "1) If a subgoal completes, ... until the robot drops it" (L137-143). This is an artifact of the AI2THOR environment, and may not generally be applicable in other simulators and in the real world.]** \
> **A:** Thank you for pointing it out. We agree with you that we have some assumptions based on the artifact of the AI2THOR environment. The assumptions include:
> 1) If a subgoal completes, the object and its corresponding receptacle objects must be displayed in the current visual frame, and most likely in adjacent frames.
> 2) If the robot agent acts "Pickup an object", the object appears in visual observation until the robot drops it.
>
> However, the assumptions do not affect the concept learning performance and could be applied to real environments. The only requirement of our concept learner is that the object is not fully occluded and can be detected as an object proposal. It is reasonable because humans also use similar steps and observations to solve tasks (also, when a robot picks up an object, we could make the object appear in visual observation).
>
> The discontinuous motion problem is a limitation of ALFRED that the action execution is not feasible, i.e, picking up an object by only one command without robot manipulation. However, the low-level control task and the current embodied instruction following task are orthogonal, which means the two tasks can still be decoupled in real-world scenarios, while our model focus on instruction following.

---

### Author Response · Authors · 2022-08-27
**General Response: Contributions and New Experiments (Revised PDF)**

We sincerely appreciate all reviewers’ time and efforts in reviewing our paper. We are glad to find that AC and reviewers generally recognized our contributions:
* **Model.** Novel and interesting approach [rRFK, Zyvv, XMVt]; Convincing motivation [TjRQ]; Might be broadly useful to the embodied AI and robotics community [XMVt]
* **Experiments.** Promising/impressive results [rRFK, Zyvv, dHJH, XMVt, TjRQ]; High generalizability [rRFK, Zyvv, dHJH, TjRQ]; Well-analyzed [TjRQ].
* **Writing.** Understandable, well-structured, supported by nice and helpful figures [rRFK, TjRQ].


And we also thank all reviewers for their insightful and constructive suggestions, which help a lot in further improving our paper. In addition to the pointwise responses below, we summarize supporting experiments added in the rebuttal according to reviewers’ suggestions.

**New Experiments and Contents**

We have updated our manuscript and appendix to include the following experiments and contents:
1. We replace the ECL-Oracle in the paper by 1) using ground truth depth; and 2) using both ground truth segmentation and depth, and report the results.
2. We report the exposure frequency v.s. concept learning accuracy of object classes, and show that they are positively correlated.
3. We report the accuracy of the instruction parser.
4. We have added some references to back up our claims.
5. We rephrase our paper to avoid overclaiming and make our paper more readable.

An updated manuscript is attached. We hope our pointwise responses below could clarify all reviewers’ confusion and alleviate all concerns. We thank all reviewers’ time again. Thank you!

Best, \
Authors

---

### Meta-Review · Area_Chair_rRFK · 2022-08-12

**Recommendation:** Accept (Poster)
**Confidence:** 4

**Metareview:**

The novel interesting approach for self-supervised learning of object-centric concepts in challenging 3D environments is recognized and appreciated. Also the presented results in ALFRED are promising, especially the generalization to unseen tasks. Moreover the paper is presented in an understandable, well-structured way, supported by nice and helpful figures.

Nevertheless, there are also some concerns raised. Mainly on clarifications on the training scheme, the underlying assumptions and how tight the approach is tailored to ALRED only. Please see the following proposed action items based on those (more details in the corresponding reviews).

Proposal of action items:
- Clarifying the training procedures and details, the usage and accuracy of the pre-trained models
- Discussing the assumptions and transfer to other simulators/real world, or adding an additional experiment
- Clarifying the usage of ‘labels’ in ALFRED
- L231 the same objects are referenced as hard to learn, but were easy to learn before?
- Adding cognitive science literature to support the human-like statements
- Rethinking the claims and wording with respect to the method, assumptions, and usage of pre-trained models

-----

I’d like to thank all the reviewers and the authors for constructive discussions and provided updates. There is a consensus that the presented work is tackling a challenging and important problem in the field of embodied AI, and an effective and promising framework is presented. The overall recommendations are to accept the paper.



**Best Paper Nomination:**

No

---

> ### Author Response · Authors · 2022-08-27
> **Response to Meta Reviewer #rRFK**
>
> Dear Meta Reviewer,
>
> Thanks for the time and effort in handling our manuscript. We appreciate the meta-reviewer's summary of strengths and weaknesses, and the serious jobs made by all the reviewers.
>
> **Q1. [Clarifying the training procedures and details, the usage and accuracy of the pre-trained models.]** \
> **A:** Thanks. We have clarified point-by-point details about training and the usage of the pretrained models to each reviewer.
>
>
> **Q2. [Discussing the assumptions and transfer to other simulators/real world, or adding an additional experiment.]** \
> **A:** Thanks. We have discussed the assumptions to transfer to other simulators or real-world scenarios as follows.
>
> Real robotics applications have been one of the longstanding motivations for this work and have been carefully considered by the authors in the design of ALFRED. When generalizing our model to real-world scenarios:
> - The instruction parser is supervisedly trained, and can be directly employed in the real world.
> - The embodied concept learner should work when there are real-world demonstrations. Currently, we have some assumptions based on the artifact of the AI2THOR environment. However, the assumptions do not affect the concept learning performance and could be applied to real environments. (The only requirement of our concept learner is that the object is not fully occluded and could be detected as an object proposal. In this way, our concept grounding does not depend on specific environments or the object's locations.)
> - Unsupervised depth and mapping are well-studied problems in the real world. We see this as a reasonable assumption for the time being.
> - Still, there are some limitations in ALFRED that the action execution is not feasible, i.e, picking up an object by only one command without robot manipulation. However, the low-level control task and the current embodied instruction following task are orthogonal, which means the two tasks can still be decoupled in real-world scenarios, while our model focus on instruction following.
>
> However, it's really challenging for a robot to perform instruction following in an unseen real-world environment, even in a simulated environment (test unseen success rate of only 23.6% even in our oracle model). To this end, ALFRED simplifies the hard problem of making meaningful progress through tight integration between visual perception, language instruction, and robotic navigation and manipulation. To the best of our knowledge, no other benchmarks contain language instructions in an interactive 3D environment with visual observation and navigation. Also, the prior works [5,6,20,21] towards this goal evaluate only on the ALFRED benchmark. As the field progresses, we are confident more works and benchmarks will be introduced, and we will take it as our future research direction.
>
>
>
> **Q3. [Clarifying the usage of ‘labels’ in ALFRED.]** \
> **A:** Thanks. We have clarified the usage of ‘labels’ in ALFRED to reviewer #dHJH. In summary, in addition to expert demonstrations (language instructions), we use two pretrained models in our framework but without using semantic or depth labels in ALFRED.
>
>
> **Q4. [L231 the same objects are referenced as hard to learn, but were easy to learn before?]** \
> **A:** Thanks. We have fixed the typo and updated our manuscript.
>
>
> **Q5. [Adding cognitive science literature to support the human-like statements]** \
> **A:** Thanks. As suggested by Reviewer #XMVt, we have added cognitive science literature to back up the claims in our paper.
>
>
> **Q6. [Rethinking the claims and wording with respect to the method, assumptions, and usage of pre-trained models]** \
> **A:** Thanks. We have toned down our claim and made it clearer in our updated manuscript. We also have clarified point-by-point details about our claims and the usage of the pretrained models to each reviewer. All the details will be delivered in our paper to avoid overclaiming and make our paper more readable.
>
> \
> An updated manuscript is attached. Thanks again for your time and effort!
>
>
> Best, \
> Authors